# Pain and other complications of pelvic mesh: a systematic review of qualitative studies and thematic synthesis of women's accounts

Amanda C de C Williams ![ORCID],[1] Mary Lodato,[2] Honor McGrigor[3]

[1]Psychology and Language Sciences, UCL, London, UK
[2]Independent Researcher, Kettering, UK
[3]Psychology & Language Sciences, UCL, London, UK

**Correspondence to**
Dr Amanda C de C Williams; amanda.williams@ucl.ac.uk

## ABSTRACT

**Objectives** Synthesis of the experience of women with pain from pelvic or vaginal mesh or its removal, to identify pain-related problems and to formulate psychological aspects of pain.

**Design** Systematic review and thematic analysis of qualitative studies of pain from pelvic or vaginal mesh, or mesh removal, in women over 18 years, using individual interviews, focus groups, free text, or written or oral contributions to formal enquiries.

**Data sources** Medline, Embase and PsycINFO, from inception to 26 April 2023.

**Eligibility criteria** Qualitative studies of pain and other symptoms from pelvic or vaginal mesh or its removal; adults; no language restriction.

**Data extraction and synthesis** Line-by-line coding of participant quotations and study author statements by one author to provide codes that were applied to half the studies by another author and differences resolved by discussion. Codes were grouped into subthemes and themes by both authors, then scrutinised and discussed by a focus group of mesh-injured women for omissions, emphasis and coherence. Studies were appraised using an amalgamation of the CASP and COREQ tools.

**Results** 2292 search results produced 9 eligible studies, with 7–752 participants, a total of around 2000. Four recruited patients, four totally or partially from mesh advocacy groups, and two were national enquiries (UK and Australia). Four major themes were as follows: broken body, broken mind; distrust of doctors and the medical industry; broken life and keeping going—a changed future. Psychological content mainly concerned the loss of trust in medical care, leaving women unsupported in facing an uncertain future. Mesh-injured women strongly endorsed the findings.

**Conclusions** Pain and other problems associated with pelvic mesh are profound and far-reaching for women affected. Worse, they feel subject to continued gaslighting, including denial of their mesh-related problems and dismissal of their concerns about continued mesh insertion.

**PROSPERO registration number** CRD42022330527.

## STRENGTHS AND LIMITATIONS OF THIS STUDY

⇒ Involvement of women with mesh complications in reviewing and discussing findings contributes to confidence in their scope and content.
⇒ Contributing women with mesh complications were all from the UK, but ideally would have been from a wider range of backgrounds.
⇒ Despite no language restriction, the studies reviewed had low ethnic diversity and were all from high-income English-speaking countries, limiting applicability of review findings.

## INTRODUCTION

The use of synthetic mesh to repair pelvic organ prolapse (POP) or for stress urinary incontinence (SUI), both relatively common conditions in women, proliferated under weak regulation and without clinical trials, drawing instead on successful use of mesh in hernia repair.[1] An early review[2] listed complications including erosion, fistula, infection, chronic pain and dyspareunia and cited the 2008 US Food and Drug Administration recommendations to 'be vigilant for potential adverse events' and to inform patients of possible serious complications. Two small systematic reviews[3 4] on mesh surgery for POP and SUI found adverse effects were poorly recorded and follow-up inadequate so both recommended a conservative approach. Guidelines published between 2015 and 2017 reported weak stakeholder (particularly patient) involvement, and inadequate declaration of competing interests.[5]

Pelvic mesh insertion was halted in the UK in 2018 and US in 2019. Reports of serious problems, particularly pain, reached public attention[6 7], generating mass legal action in the USA. Complication rates from UK hospital data were estimated as 9.8%–12.8% over 5 years of follow-up.[8] Several studies of women who had undergone mesh insertion[9–12] suggested that they were poorly informed about adverse effects or alternative treatments, with internet information of variable quality.[13] Women who developed

problems with mesh often had considerable difficulty convincing doctors of their symptoms and that mesh was the cause or obtaining adequate care.[14 15] Formal enquiries in the UK (The Cumberlege Report),[16] Scotland[17] and Australia[18] recorded widespread and severe distress and substantial shortcomings in care. A systematic review of mesh complications[19] found only one prospective study, and very varied outcomes of pain and other symptoms and little on quality of life. A qualitative systematic review[14] described how discounting of women's experience compounded the psychological harm from mesh.

More recent studies and rich material from national enquiries provide data for a larger and more critical meta-synthesis of qualitative studies. A particular focus here was the relationship of pain to mesh-associated disabilities: the standard model of pain, developed in musculoskeletal patient populations, identifies fears of increased pain or damage as the basis of extensive activity avoidance that constitutes a disability,[20 21] but the extent to which this applies to visceral pain is uncertain.

## METHODS

This systematic review was registered with the International Prospective Register of Systematic Reviews (PROSPERO CRD42022330527). In preparation for the review, the researchers discussed mesh-related pelvic pain and key literature with clinicians involved in treatment and consulted an academic librarian about search terms and databases.

### Search strategy

On 24 October 2022, a comprehensive literature search of Medline, Embase and PsycINFO was conducted and updated by repeating it on 26 April 2023 (see online supplemental data for search terms). Following each search, citation chaining was used.

### Inclusion and exclusion criteria

The inclusion criteria were (1) qualitative research on pain from pelvic or vaginal mesh, or pain after mesh removal; (2) in adults (18 and over) and (3) in peer-reviewed journals or publicly available PhD theses. No limitations were placed on language or date of publication.

### Study selection

Records from the searches and citation chaining were exported to Endnote V.X9.3.3 and deduplicated using automated than manual methods. The remaining records were screened (by HM) on title and abstract, and ineligible records were removed; the lead researcher (ACdCW) checked a 5% random sample of these rejected records. Possible records were retrieved as full texts, read by both researchers independently to decide on inclusion or reasons for exclusion.

### Evaluation of studies

The characteristics of studies were appraised using an amalgamation of the Critical Appraisal Skills Programme (CASP)[22] and Consolidated Criteria for Reporting Qualitative Studies (COREQ)[23] quality assessment tools (see online supplemental file 1) from which essentially similar items had been removed. Both researchers rated the included studies independently and discussed their ratings to achieve consensus.

### Data synthesis

The data were treated according to Thomas and Harden's thematic synthesis method.[24] Using NVivo V.12 1.6.1, one researcher (HM) read all the texts, generated initial codes using inductive methods, then coded results (including direct quotations) and discussions of included studies, line by line. These codes were used by the other researcher (ACdCW) in five of the nine studies. This enabled collapse of many codes to produce a more compact set. Both researchers generated subthemes and themes from these codes.

### Positionality and reflexivity

Given the subjective bias that affects data analysis, we provide the following information for readers to consider. ACdCW is an academic and clinical psychologist with over 35 years' experience of in clinical and research work in chronic pain, including chronic pelvic pain. She questions the fit of the fear and avoidance model in research on or clinical formulation of visceral pains. HM is a research psychology assistant, with experience in qualitative methods but not in pain. Both researchers aimed for reflexive processing of data, considering at multiple points whether and how their beliefs and concerns might influence their decisions. Neither has personal experience of chronic pelvic pain or mesh.

### Patient and public involvement

Following analysis of themes and subthemes, the views of women with mesh complications were sought. An advertisement was circulated via a member of the project's patient and charity advisory board (PCAB) to an author who is a leading member of an advocacy group that campaigns for better services and care for women with mesh complications; these collaborators also distributed information and collected consent from women who volunteered to take part. Participating women were provided with the full thematic analysis results with 2 days to read them before taking part in an online meeting, hosted by the PCAB member for pelvic mesh, at which results were discussed. Notes were taken by the first author of the paper, with a full verbatim transcription of a recording of the meeting by the PCAB member. Women were recompensed for their contributions.

## RESULTS

The 3232 records from the searches and citation chaining were reduced to 2292 by automatic deduplication (653 records) then manual removal (287 records). Screening of these titles and abstracts removed 2273 ineligible

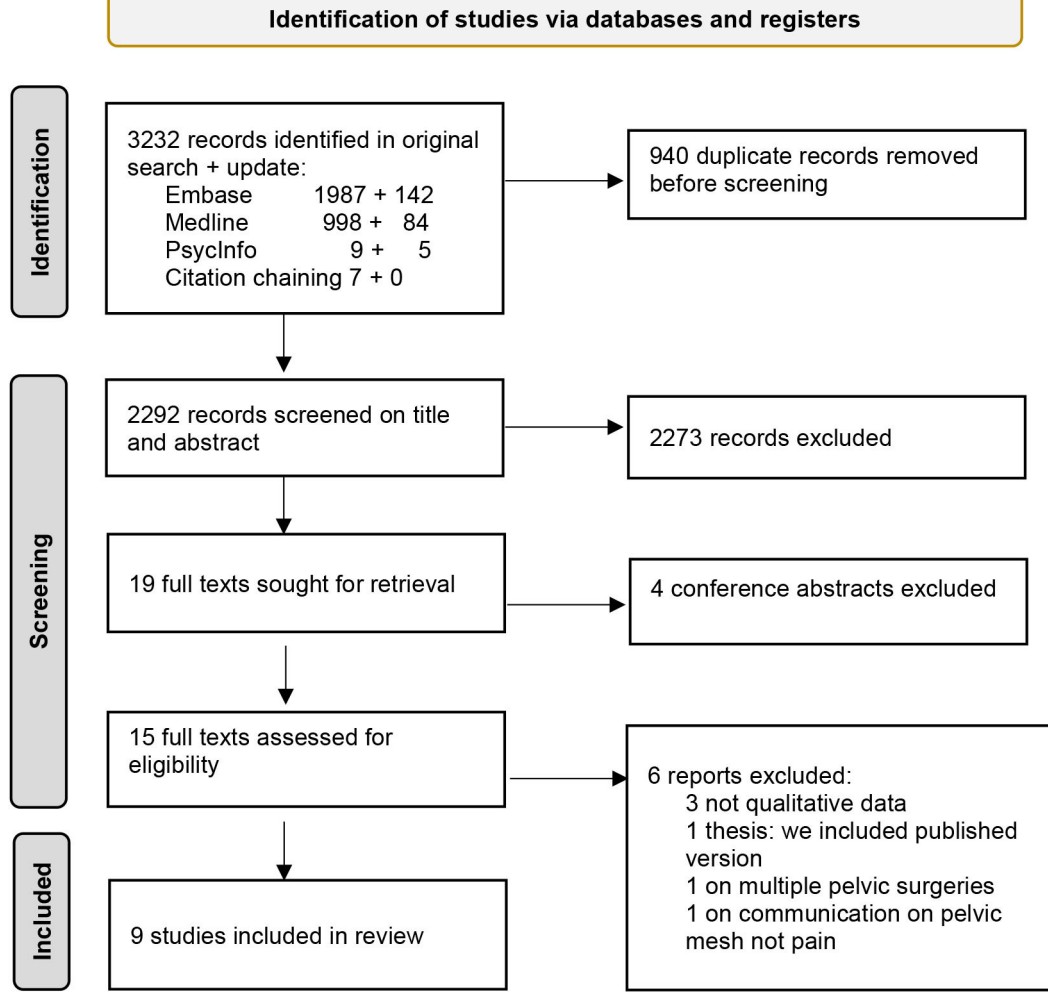

**Figure 1**

records; there were no disagreements on the sample of rejected records. The remaining records were discussed, resulting in full-text retrieval of 19 potentially eligible qualitative studies and a further rejection of 4 conference abstracts. Of the 15 remaining, 9 were included (see figure 1, Preferred Reporting Items for Systematic Reviews and Meta-Analyses diagram). The excluded studies were two government reports[17 25] and the transcript of a television documentary[6] that were not designed or reported in the form of qualitative data; one master's thesis[26] for which we included the published paper[27]; one study of multiple pelvic surgeries among which accounts of mesh were not distinguishable[28] and one use of written evidence to a government enquiry to study women's accounts of communication about mesh, not addressing pain directly.[29]

### Characteristics of studies and participants

The included studies are described in table 1. Four studies were conducted in the UK,[16 30–32] two in the USA,[33 34] two in New Zealand[27 35] and one in Australia.[36] Participant numbers varied from 7 to 752, with a total across the 9 studies approaching (and possibly exceeding) 2000. Four studies were recruited from patient populations exclusively[27 31 33 34] and one partially[32]; four from advocacy groups for affected women, two exclusively[30 35] and two partially[16 32]; and two drew on material from national enquiries.[16 36] Where non-patient participants such as carers and clinicians also provided material for the report,[16 36] we used only submissions from affected women or representatives of mesh advocacy groups. Five studies used semistructured[27 30 32 34] or structured[33] interviews; two drew from free text that supplemented questionnaire responses[31] or national inquiry[36]; one used free text emailed responses[35] and one used transcribed oral responses from inquiry hearings and written responses to drafts of the report.[16]

Seven studies provided information on age, five[27 30 31 33 34] with a mean age in the 50ss, and range from 20 to 80s; the other two[32 35] provided ranges from the thirties into the seventies. Only three provided information on ethnicity, all majority or entirely white (European, non-Hispanic),[27 32 34] but it is likely that the other studies were similar in this respect. Four studies were recruited from clinical populations[27 31 33 34] and one partially so[32]; two through social media support groups[30 35]; and two issued open invitations to contribute to national inquiries.[16 36] As

**Table 1** Characteristics of included studies

| Author, date of publication, author details, financial interests | Title | Research focus | Recruitment | Sample size | Data collection method |
|---|---|---|---|---|---|
| Brown 2020 F nurse with lived experience of mesh. No financial interests.[27] | The experiences of seven women living with pelvic surgical mesh complications | Lived experience | Women attending physiotherapy | 7 | Semistructured interview |
| Cumberlege (chair) 2020; F politician and life peer; panel M physician, M communication consultant, F secretary. No financial interests.[16] | First do no harm: the report of the Independent Medicines and Medical Devices Safety Review | Adverse experiences, information useful for making recommendations | Mesh patient groups, affected individuals including carers | Unclear: >100 | Independent inquiry: patient engagement events, feedback on drafts |
| | Annex J: Personal testimonies | | | Unclear: >10 | Written |
| | Annex K: Oral hearing transcripts | | | 5 women with mesh, 1 carer, 12 mesh group reps; 10 clinicians | Oral accounts in hearings; transcribed, plus one letter |
| Dibb et al, 2023, all 3 F health researchers. No financial interests.[30] | When things go wrong: experiences of vaginal mesh complications | Complications of mesh and their impact | Mesh support group on social media | 18 | Semistructured interview |
| Dunn et al, 2014, all 7 F doctors or nurses mainly in urogynaecology. No financial interests.[33] | Changed women: the long-term impact of vaginal mesh complications | Women's experience of mesh complications after specialist care | Urogynaecology clinic for mesh complications | 84 | Structured telephone interview |
| Huntington et al, 2019, 2 F health researchers, 1 F mesh group advocate. No declaration of interest.[35] | The loss of a life well lived: a qualitative study exploring the impact of surgical mesh implants on the lives of a group of New Zealand women | Impact of mesh complications | Mesh support group through health advocate | 23 | Emailed account |
| Izett-Kay et al, 2020, 5 M surgeons, 1 F consultant, 1 F medical researcher. 3 disclosed financial interests.[31] | 'What research was carried out on this vaginal mesh?' Health-related concerns in women following mesh-augmented prolapse surgery: a thematic analysis | Health problems after mesh insertion | Patients of 5 surgeons | 752 | Free text responses on written/online questionnaire |
| McKinlay and Oxlad 2022, 2 F health researchers. No financial interests[36] | 'I have no life and neither do the ones watching me suffer': women's experiences of transvaginal mesh implant surgery | Impact of mesh, taking biopsychosocial perspective | Written submissions from national inquiry into mesh | 153 | Free text from submissions |
| Toye et al, 2023, 3 health researchers, 1 M surgeon. No financial interests.[32] | The experience of women reporting damage from vaginal mesh: a reflexive thematic analysis | Explore and understand the experience of living with complications of mesh | Women being treated for urogynaecological conditions through healthcare, advocacy groups, advertisement, snowball sampling | 15 | Semistructured interviews, telephone or video call |
| Uberoi et al, 2021,1 M and 2 F surgeons, 1 M urologist, 2 F researchers. No financial interests.[34] | Listening to women: a qualitative analysis of experiences after complications from mesh mid-urethral sling surgery | Understand women's experiences after mesh revision | Patients of 3 surgeons | 19 | Semistructured interviews and focus groups |

F female, M male.

## Box 1   Thematic analysis

*The main themes are in **bold and underlined.** Subthemes use as heading a quotation from a mesh-affected woman in one of the studies Subthemes show constituent codes, the most frequently occurring in **bold**, and the least frequent in grey. Positive comments that belong in the code are prefixed and suffixed by a '+'.*

**Broken body, broken mind**
'my life is never going to be the same'
⇒ **this is my life now, 'new normal'**
⇒ **permanent problem, ruined life, reduced quality of life**
⇒ impact on identity, changed as a person, perspective changed, life on hold, lack of fulfilment
⇒ **chronic pain,** descriptions of extreme pain, lower back pain
⇒ not able to function, lost trust in body, feeling broken
⇒ grief, loss, feeling robbed
⇒ bladder problems, pain, dysfunction, discharge, repeated infections, abnormal bleeding, bowel problems, incontinence; practical issues associated with bleeding, discharge, incontinence
⇒ comorbidities, cascading health issues, fatigue, tiredness, consequences of medication, sleep disruption, weight gain
⇒ shame, embarrassment, loss of confidence, impact on self-esteem, hopelessness
'I can't achieve very much'
⇒ **being or doing less than before surgery**
⇒ activity and physical limitations, loss of mobility, daily difficulties, limitations on daily life, worsening after activity, not being able to sit or stand, not being able to do housework, restriction on travel
⇒ disability, feeling like a burden, loss of independence
'It has left me feeling lost, extremely anxious'
⇒ **anxiety, mental health affected, distress, suicidal feelings,** depression, feelings of frustration and anger, emotional volatility, 'emotional wreck', guilt, self-blame, unhealthy coping mechanisms, for example, alcohol
⇒ having psychological treatment, counselling, therapy
'You can't have that [sexual] relationship with someone screaming in pain'
⇒ loss of intimacy, impact on sex affecting relationship, penetrative sex as impossible, dyspareunia
⇒ generic sexual problems, avoidance of sex
⇒ partner feels mesh during sex [validating]
⇒ **Linked to both broken body and distrust of doctors**
'I am frightened if I take it out; I am frightened if I don't'
⇒ **fear of future problems and future surgery,** uncertain future
⇒ multiple operations or hospitalisation to fix subsequent problems, wishing for mesh removal, remaining mesh, mesh as alien, foreign in the body, mesh erosion

**Distrust of doctors and medical industry**
'She suggested that it was such an easy fix'
⇒ **feeling misinformed about some or all risks**, **not knowing, being lied to,** 'quick fix', benefits overstated, lack of informed consent and informed choice, feeling 'sold' on mesh, regret surgery
⇒ lacked or wanted more discussion of alternatives to surgery
⇒ preoperative expectations of improvement after surgery, recovery taking longer/being harder than expected
⇒ feeling dehumanised, 'human guinea pigs'
'you're the only person I've seen who is complaining and thinking you have problems'
⇒ **dismissal of patient concerns, 'it's all in your head', 'there is nothing wrong with you'**
⇒ doctors not taking responsibility for the problem, doctors not giving attention needed, lack of empathy, insensitive medical professionals

Continued

## Box 1   Continued

⇒ doctors blaming women
'I trusted fully all I was told'/'I was in a very vulnerable position and felt unable to say no.'
⇒ **trust lost**
⇒ **should not have put trust in doctor, importance of patient–provider relationship, power dynamic in patient–doctor relationship**
'All that I ask is honesty'
⇒ health system as understandably fallible—no time, doctors as people—etc; wanting more transparency, wanting acknowledgement of what has happened
⇒ adverse event need to be logged, problems with mesh described as 'unusual' by doctors, medical professionals needing more education on mesh
⇒ looking for information.+positive interactions with medical professionals+
'I have beaten cancer, but mesh [has] beaten me'
⇒ **victims of mesh, medical companies**
⇒ trauma, medical trauma, PTSD, mesh should be banned
⇒ danger—potentially fatal
⇒ litigation, financial compensation desired, battle to obtain financial compensation

**Broken life**
'My children needed their mother back'
⇒ **relationship with grandchildren and children affected**, impact on family, relationship with partner affected, dynamic changed
⇒ +family is reason for living, my family and friends keep me going; support from partner+
⇒ unsupported by partner, breakdown of relationship with partner
'people get bored with it, and they're not interested, and you sort of get dropped'
⇒ **isolation, loneliness**
⇒ **not being listened to, not being believed,** suffering in silence, people don't want to hear about it
⇒ social relationships and friendships affected, social life affected, preventing new potential relationships
'I am unable to work … I miss being able to contribute'.
⇒ impact on career, loss of job, having to take time off work, financial burden of being able to work
⇒ financial burden of treatment, medication, supplies

**Keeping going —'a changed future'**
⇒ wanting to help others, concern for others with mesh complications, being able to relate to others with the same condition
⇒ +successful mesh experience+, how women judge their surgery, what is judged as success
⇒ +positives that have come from vaginal mesh; positive hopes for the future+
⇒ vaginal mesh community being upsetting

far as could be ascertained, two studies recruited women with mesh still in place,[31 35] and the remainder recruited a mix of women with mesh in place, mesh partially removed, mesh fully removed or having had unspecified revision surgery.[34]

### Aims and methods of included studies

Information collected using the combined COREQ/CASP form is summarised here (see online supplemental file 1 for detail). Six studies aimed to describe the experience of

women with mesh complications, five[27 30 32 34 35] on the basis that it had been inadequately addressed in the literature and one to follow up 'optimised' specialist treatment of complications.[33] The UK inquiry[16] also aimed to recruit women with mesh-related complications. The remaining two papers aimed rather to capture varied experiences from women after mesh surgery: one using written inquiry data to explore experience 'through a biopsychosocial lens',[36] the other to explore 'health-related issues' in a more 'balanced' way than those that focused on mesh complications.[31] It should be noted that the conflicts of interest for this latter study disclose that three of the eight authors had associations with mesh producers.

Two studies[34 36] combined deductive and inductive approaches; four were inductive,[27 31 32 35] one implied an inductive approach but some themes rather closely resembled question topics,[30] one simply described 'low level inference' in its analysis[33] and the inquiry[16] took a transparent approach to reporting but did no formal qualitative analysis. Two studies[27 32] and one inquiry[16] discussed and revised their findings with the help of participants.

Some methodological details are of interest. Of those studies that interviewed participants, five used female interviewers[27 30 32 34 35] and another probably did so but was unclear[33]; one inquiry[16] had a female chair with two male panel members and a female secretary. One study in which the researcher herself had experience of mesh surgery disclosed this to participants[27]; her paper discusses reflexivity and bias at some length. In another study,[35] one of the researchers was a health advocate who belonged to the online mesh group from which participants were recruited but did not explore the implications of this for data or analysis. Where participants were patients treated by authors,[31 33] possibly,[34] there is a lack of transparency about potential effects on recruitment, data collection and data analysis. The inquiry[16] carried out in person by a panel of experts in healthcare and in public enquiry processes describes its efforts to ensure transparency and openness, and its independence from governmental or industry influence.

### Thematic synthesis

Initial coding of content of results and discussions of all studies, drawing both on directly reported participant comments and on researcher commentary, provided 101 codes. These were collapsed and grouped by the researchers collaboratively. Subthemes were named as far as possible using quotations. The final themes and subthemes, with content, are shown in box 1, and the studies contributing in table 2.

There was a strong sense of double betrayal in women's accounts: feeling misled about the likely success and possible harms of the original mesh insertion surgery, and not offered alternative nonsurgical interventions; then not being believed or treated with adequate care when they reported problems post-surgically. The main themes, broken body, broken mind and broken life reflected in some detail the extent and severity of adverse effects from the mesh. Chronic pain was prominent, as was incontinence and other bladder, bowel and wider health problems. These were interwoven with frustration at the limitations imposed by pain and incontinence, and a powerful sense of loss of family and social relationships. Closely related to both of these was the sense of distrust of doctors and the medical industry engendered by the original decision about mesh insertion and by the uncaring response to symptoms and problems that followed. Some women felt deliberately misled by doctors, mostly surgeons, but many contextualised their experience in lack of knowledge and information among the medical profession, and in their fallibility. Despite a few positive comments about interactions with doctors, the avoidability of the disastrous experience left many women bitter about having agreed to mesh insertion.

A fourth and somewhat separate theme concerned adjustment to the situation, keeping going—a changed future. This contained ways that women had made meaning from their experience, such as activism on behalf of and advocacy for women with mesh complications. Some of the comments about positive experience were apparently spontaneous, but others were elicited by leading questions.[30]

### Review by women with experience of mesh

Seven women with complications of mesh took part in the online meeting, plus another who facilitated the meeting arrangements and transcribed the meeting content. Some of the women had undergone mesh removal and others had not; for most, symptoms persisted or had worsened. Their overall impression was that the themes were familiar to them and described their experiences; no major areas were raised that were missing, and they did not think that the negativity of the themes was unrepresentative. The women endorsed in particular anger about the original surgery, about treatment, and about subsequently about not being believed when they presented with complications, and still not being taken seriously when they sought medical help, even unrelated to mesh. This they associated with a general dismissal of women's health problems, and the defensiveness of medicine when the possibility of iatrogenic harm was raised. All these contributed to a loss of trust and confidence in the institution of medicine. There was some concern that the themes did not adequately articulate the moral wrong of having been 'mutilated by the medical industry'.

Further, they were concerned about lack of accountability for money spent on the recommendations of the UK Cumberlege Inquiry[16] that had been accepted by the government. They were sceptical about the adequacy of staff training in the nine specialist centres now responsible for the care of women with mesh complications and about surgeons removing mesh who had previously been committed to implanting it. Women had hoped for the establishment of holistic and integrated care of the sort offered in some cancer services but experienced rather a fragmentary service, little follow-up after removal. They

**Table 2** Coverage of themes and subthemes by studies

| | Brown[27] | Cumberlege[16] | Dibb et al[30] | Dunn et al[33] | Huntington et al[35] | Izett-Kay et al[31] | McKinlay and Oxlad[36] | Toye et al[32] | Uberoi et al[34] |
|---|---|---|---|---|---|---|---|---|---|
| **Broken body, broken mind** | | | | | | | | | |
| 'My life is never going to be the same' | y | y | y | y | y | y | y | y | y |
| 'I can't achieve very much' | y | y | y | y | y | y (+) | y | y | y |
| 'It has left me feeling lost, extremely anxious' | – | y | y | y | y | y | y | y | y |
| 'You can't have that (sexual) relationship with someone screaming in pain' | y | y | y | y | y | – | y | y | Y |
| **Linked to both broken body and distrust of doctors** | | | | | | | | | |
| 'I am frightened if I take it out; I am frightened if I don't' | y | y | y | y | y | y | y | y | Y |
| **Distrust of doctors and medical industry** | | | | | | | | | |
| 'She suggested that it was such an easy fix' | y | y | y | y | y | y | y | y | Y |
| 'You're the only person I've seen who is complaining and thinking you have problems' | y | y | y | y | y | y | y | y | Y |
| 'I trusted fully all I was told' | y | y | y | y | y | – | – | y | Y |
| 'All that I ask is honesty' | y | y (+) | – | y | y | y (+) | – | y (+) | y (+) |
| 'I have beaten cancer, but mesh (has) beaten me' | y | y | y | y | y | y | y | y | Y |
| **Broken life** | | | | | | | | | |
| 'My children needed their mother back' | y | y | y | – | y | – | y | y | Y |
| 'People get bored with it, and they're not interested, and you sort of get dropped' | y | – | y | y | y | – | y | y | Y |
| - 'I am unable to work … I miss being able to contribute' | y | y | – | y | y | – | y | y | – |
| Keeping going—'a changed future' | – | y | y | y | y | y | – | y | y (+) |

+ =+ = positive aspects, for example, positive interactions with medical professionals.

were aware of continued pressure from some surgeons to lift the current ban on pelvic mesh. Alongside these deeply distressing experiences, women also gave credit to the GPs and surgeons they had encountered who were concerned and willing to listen and learn. Several women had pursued medicolegal cases, but some necessary medical examinations had not been performed by surgeons who were experts in mesh-related problems. Other women had felt that internal examinations and psychiatric interviews were requested in order to discourage their litigation.

Women were curious about, and some were critical of, research methodologies; they emphasised how important it was to know what questions were asked when analysing the answers that provided data for qualitative studies: that they might have been designed to elicit positive responses about mesh. They also raised the issue of vested interests of some clinician researchers who benefited from ongoing relationships with mesh companies, and other hidden agendas (such as lifting the ban on pelvic mesh) informing research design and findings.

## DISCUSSION

Three of the four themes were overwhelmingly negative in emotional tone; only the last theme, Keeping going—a changed future, had a more varied tone, but was extracted from fewer studies (see table 2). Nevertheless, it was endorsed by the women with mesh complications who discussed the findings and described how they directed their anger about their experience into helping other women with mesh complications, and that meeting other women with similar experiences had been hugely important, far beyond validation of their current difficulties.

The destructive impact of mesh complications, and in some cases further impact of mesh removal, was evident across somatic, emotional, family, social and vocational domains of life, with a deep sense of irreversible loss. The two themes Broken body, broken mind and Broken life, attest to widespread adverse effects of the pelvic mesh; one of the women reviewing the results commented that her sense of being female had been destroyed by the experience. When major health problems occur, people often ask themselves why it happened to them, and whether it could have been prevented.[37] This provided the basis for the third main theme, Distrust of doctors and medical industry. Not only did women feel misinformed about the options for surgery when they first presented with prolapse or stress incontinence, but also that the risks of surgery had not been known or communicated to them. Worse, when they experienced complications of mesh, their symptoms and distress were frequently dismissed, even denied, by doctors. A study of surgeons' reasons for continuing mesh insertion showed a focus on repairing anatomy rather than patient experience,[38] deflection of blame[39] and a lack of evidence, since no denominator of total mesh insertions existed for estimating harms.[40] While

doctors themselves had been inadequately informed of risks by an industry that showed little interest in accurate estimation of outcomes,[40] many women also experienced their doctors as being uninterested in the outcomes of surgery they had performed, or (in primary care) recommended. The belief that women imagine, exaggerate, and fail to manage their symptoms persists throughout healthcare,[41] and these women felt additionally disqualified because their problems were iatrogenic.

From a psychological point of view, the problem of pain was overshadowed by many other mesh-related symptoms and losses; there was no evidence that as in the standard model, women's disabilities were the outcome of unwarranted fears for their health and overcautious decisions about activity, although of course data were not collected specifically to test this model. It would be a serious error to interpret women's accounts as catastrophic overestimation of threat from innocuous events.[42] It is not possible to assert on the basis of these findings that pain was a predominant cause of disability, but it was a common reason for seeking medical help among women experiencing complications.

### Strengths and limitations

There are some limitations to this meta-synthesis that arise from characteristics of the studies included. Despite a search without a language limit, studies were all from high-income English-speaking countries, but enthusiasm for mesh insertion persists in high-income and middle-income countries.[43] Disclosure of interests was inadequate in several studies, including those that declared some, and there was a general lack of reflexivity from clinician researchers about how their training and outlook might affect their questions and the answers they obtained, particularly when interviewees were their own patients. There was also little discussion in studies of the problems of researcher-selected or self-selected participants providing a limited range of concerns, especially where samples were small. Themes not represented in individual studies could not appear in the meta-synthesis: however, the women with mesh complications who were consulted about the meta-synthesis findings did not identify any major gaps. The women who contributed were all from the UK: a wider sample would have been desirable. There is always subjectivity in coding and construction of themes from codes, and a statement of reflexivity and positionality does not remove subjectivity, only allows readers to judge bias for themselves without a formal measure.

We have moderate to high confidence in our findings. Using the CERQual categories of methodological limitations, coherence of findings, adequacy of data and relevance of findings,[44] we note methodological shortcomings in not having a larger team to contribute to the analysis, a weakness only partly mitigated by involving a group of mesh-injured women, and although samples in several studies were large and data-rich in most, some populations are poorly represented in the nine included studies, compromising data adequacy.

## Implications of the review

The industry implications have been addressed in the national enquiries,[16 36] and by mass legal action in the USA: permission to extend the use of mesh was far too easily granted,[40 45] and systematic reporting of adverse effects was weak or absent or relied on legal records.[46] The clinical shortcomings are summarised as lack of postmarketing surveillance, poor understanding of the pelvic floor and pelvic-floor-related disorders and inadequate medical training of non-mesh management of POP and SUI.[45] Available information on the internet, when studied in 2019, was of moderate quality,[13] but it is not known if it is better now. Development and user-testing of information resources is underway[47] but on a small scale.

Other clinical implications are not unique to mesh but concern gender bias in medicine that leads to disbelief or disregard of women's symptoms, and punitive interactions with women who challenge routine medical practice or who seek redress. The loss of trust expressed in the studies, and in the consultation with women with mesh complications, was shocking. It is only in New Zealand that governmental initiatives have addressed this directly, using restorative justice methods.[25] Lastly, the standard model of chronic pain disability as arising more in unwarranted fears than in pain itself fails to describe these findings, and application to pelvic mesh complications would only exacerbate the gaslighting of women with painful mesh complications.

**Acknowledgements** We recognise the help of our PCAB member, and the courage of the participants who took part in the discussion of results, sharing their lived expertise, their personal journeys, and their trauma, in the hope that their voices would generate a deeper understanding in the literature of the harm done by mesh in the UK. We are grateful to clinicians who gave us their time and to our librarian colleague.

**Contributors** ACdCW conceived the study; ACdCW and HM designed the study; HM conducted the searches; HM and ACdCW screened and extracted studies from the search output; HM and ACdCW completed COREQ/CASP appraisals and agreed them; HM completed initial coding of studies; ACdCW and HM combined codes into themes and subthemes; ML and ACdCW consulted women with mesh about the draft thematic synthesis; ACdCW drafted the paper, HM drafted figure 1, tables 1 and 2 and supplementary information, and ML and HM critically reviewed the paper and gave final approval. All meet criteria for authorship, and contributions from other colleagues have been acknowledged.

**Funding** Our funding is from the Advanced Pain Discovery Platform, funded by the MRC, Versus Arthritis, ESRC, BBSRC, Medical Research Foundation, Astra Zeneca and EliLilly: Grant Reference No. MR/W002426/1; Primary investigator Professor Geoff Woods, University of Cambridge.

**Disclaimer** The funding organisations had no role in the design and conduct of the study, collection or analysis of data, or preparation of the paper.

**Competing interests** All authors have completed the ICMJE disclose form. All declare no financial support from any industry for the submitted work. AW and HM declare no other relationships or activities that could appear to have influenced the submitted work; ML declares leading membership of Mesh Mavericks, an advocacy group for women with pelvic mesh complications.

**Patient and public involvement** Patients and/or the public were involved in the design, or conduct, or reporting, or dissemination plans of this research. Refer to the Methods section for further details.

**Patient consent for publication** Not applicable.

**Ethics approval** This part of the review had approval from the UCL Research Ethics Committee (ref:2182 Amendment 1).

**Provenance and peer review** Not commissioned; externally peer-reviewed.

**Data availability statement** Data are available on reasonable request.

**ORCID iD**
Amanda C de C Williams http://orcid.org/0000-0003-3761-8704

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
