## [Reviewer comments · BMJ Open]

ARTICLE DETAILS

TITLE (PROVISIONAL)	Pain and other complications of pelvic mesh: a systematic review of qualitative studies and thematic synthesis of women's accounts
AUTHORS	Williams, Amanda; Lodato, Mary; McGrigor, Honor

VERSION 1 – REVIEW

REVIEWER	Curtis, Thomas Norfolk and Norwich University Hospitals NHS Foundation Trust, Obstetrics & Gynaecology
REVIEW RETURNED	15-Mar-2024

GENERAL COMMENTS	This paper represents a valuable qualitative synthesis of women's accounts, feelings and attitudes towards mesh complications that they have experienced. It is well written and provides detailed descriptions of the methodology used, particularly in sections on the reflexivity of the authors who performed the synthesis. There is strong patient and public involvement, and the addition of contemporaneous review by stakeholder groups to ensure that the themes identified in the synthesis are truly reflective of the lived experience is commendable. My only significant criticism is in the final paragraph of the methods section, under "patient and public involvement", an author (which I have understood to be ML) is described as "a leading member of an advocacy group that campaigns for better services and care for women with mesh complications." In the declaration of interests, no declaration of interest has been made. I feel that this involvement does represent a potential competing interest worthy of declaration, and the exact involvement and name of the campaign group should be included in the declarations.
--

REVIEWER	Ahern, Susannah Monash University Department of Epidemiology and Preventive Medicine
REVIEW RETURNED	24-Mar-2024

GENERAL COMMENTS	Thank you for the opportunity to review this well written and insightful paper. The paper aims, methods and results are clearly presented. The authors have presented a detailed account of the identified papers from the literature review to provide an important context for the thematic analysis. The authors also discuss appropriately potential sources of bias both in the reviewed papers that comprised the systematic review, as well as the use of reflexivity as a method to regularly check in
---

	about how their own backgrounds and views may affect their analysis. The final identified themes are well substantiated in Table 3, and importantly extend beyond pain and physical complications from mesh to include themes related to trust, psychological impact, and ongoing impacts on sense of self. These are important messages for clinician and health system leaders. I think it is worth noting that these findings relate to the women who self-identified and were selected by clinicians to take part in this research, and as such, represent the experiences and perspectives of these women only. As the authors note, the number of articles as well as size of the sample were small, and the findings can only reflect the conclusions of this sample. I recommend this paper for publication.
--	--

VERSION 1 – AUTHOR RESPONSE

Reviewer: 1	
This paper represents a valuable qualitative synthesis of women's accounts, feelings and attitudes towards mesh complications that they have experienced. It is well written and provides detailed descriptions of the methodology used, particularly in sections on the reflexivity of the authors who performed the synthesis. There is strong patient and public involvement, and the addition of contemporaneous review by stakeholder groups to ensure that the themes identified in the synthesis are truly reflective of the lived experience is commendable. My only significant criticism is in the final paragraph of the methods section, under "patient and public involvement", an author (which I have understood to be ML) is described as "a leading member of an advocacy group that campaigns for better services and care for women with mesh complications." In the declaration of interests, no declaration of interest has been made. I feel that this involvement does represent a potential competing interest worthy of declaration, and the exact involvement and name of the campaign group should be included in the declarations.	Thank you for these encouraging comments. This is a helpful criticism, and we agree that it should be disclosed. With full agreement from ML and the group Mesh Mavericks, we have submitted a revised Col form for ML, as well as revising the disclosure at the end of the paper as follows: All authors have completed the ICMJE disclose form. All declare no financial support from any industry for the submitted work. AW and HM declare no other relationships or activities that could appear to have influenced the submitted work; ML declares leading membership of Mesh Mavericks, an advocacy group for women with pelvic mesh complications.
Reviewer: 2	
Thank you for the opportunity to review this well written and insightful paper. The paper aims, methods and results are clearly presented. The authors have presented a detailed account of the identified papers from the literature review to provide an	Thank you for these positive comments on our work.

important context for the thematic analysis. The authors also discuss appropriately potential sources of bias both in the reviewed papers that comprised the systematic review, as well as the use of reflexivity as a method to regularly check in about how their own backgrounds and views may affect their analysis. The final identified themes are well substantiated in Table 3, and importantly extend beyond pain and physical complications from mesh to include themes related to trust, psychological impact, and ongoing impacts on sense of self. These are important messages for clinician and health system leaders.	
I think it is worth noting that these findings relate to the women who self-identified and were selected by clinicians to take part in this research, and as such, represent the experiences and perspectives of these women only. As the authors note, the number of articles as well as size of the sample were small, and the findings can only reflect the conclusions of this sample.	We agree that this is a limitation of substantial concern, whether women were selected by clinicians for the original studies (without any discussion of bias), or were selected formally or informally by the researcher, or self-selected in response to an advertisement or invitation. This does, of course, raise questions about who was intentionally excluded or did not contribute. We have added a sentence to the limitations, starting “There was also little discussion in studies of the problems of researcher-selected or self-selected participants...”